# EXPLAINABLE AI-BASED DYNAMIC FILTER PRUNING OF CONVOLUTIONAL NEURAL NETWORKS

## ABSTRACT

Filter pruning is one of the most effective ways to accelerate Convolutional Neural Networks (CNNs). Most of the existing works are focused on the static pruning of CNN filters. In dynamic pruning of CNN filters, existing works are based on the idea of switching between different branches of a CNN or exiting early based on the difficulty of a sample. These approaches can reduce the *average latency* of inference, but they cannot reduce the *longest-path latency* of inference. In contrast, we present a novel approach of dynamic filter pruning that utilizes explainable AI along with early coarse prediction in the intermediate layers of a CNN. This coarse prediction is performed using a simple branch that is trained to perform top-k classification. The branch either predicts the output class with high confidence, in which case, the rest of the computations are left out. Alternatively, the branch predicts the output class to be within a subset of possible output classes. After this coarse prediction, only those filters that are important for this subset of classes are utilized for further computations. The importances of filters for each output class are obtained using explainable AI. Using this architecture of dynamic pruning, we not only reduce the average latency of inference, but we can also reduce the longest-path latency of inference. Our proposed architecture for dynamic pruning can be deployed on different hardware platforms. We evaluate our approach using commonly used image classification models and datasets on CPU and GPU platforms and demonstrate speedup without significant overhead.

## 1 INTRODUCTION

Deep Neural Networks (DNNs) have achieved rapid success in many image processing applications, including image classification [1, 2, 3] image segmentation [4, 5], object detection [6], etc. The key ingredients in this success of DNN have been the usage of deeper networks and a large amount of training data. However, as the network gets deeper, the model complexity also increases rapidly. The training of DNNs can be carried out on high-performance clusters with Graphics Processing Unit (GPU) acceleration; however, for implementing these networks on hardware, the complexity of the DNNs needs to be reduced. This includes decreasing the memory requirements, energy consumption, latency, or throughput of the implementation of DNNs on the hardware. Pruning is one of the main techniques that are utilized for reducing the complexity of DNNs. DNN Pruning refers to removing the undesired parameters of a DNN that have little influence on the output of the neural network [7, 8]. This leads to fewer Multiply-Accumulate Operation (MAC) operations and fewer Number of Parameters (NPs). Pruning can be neuron pruning, filter pruning, weight pruning, and layer pruning. In neuron pruning, individual neurons are removed, i.e. , all the incoming and outgoing connections to a neuron are also removed [9]. In filter pruning, CNN filters are removed [10]. In layer pruning, some of the layers can also be pruned [11]. Weight pruning is used synonymously for unstructured pruning, where the redundant weights are set to zero. The two fundamental objectives for pruning the model are: (1) reducing the memory by lowering the NPs and reducing the latency and energy consumption of the computation by reducing MACs. These two objectives are often conflicting in a DNN because the MACs are concentrated in the lower layers of a typical image classification network and the NPs are concentrated in the higher layers of a typical image classification network.

Pruning can be structured or unstructured. In structured pruning, the filters and weights are eliminated by removing all their input and output connections, and this means that no additional compilation techniques or hardware optimization is required to obtain the gain on hardware in terms of reduction

in the size of the model or reduction in the inference time of a sample. This is because the whole model structured is changed in unstructured pruning.

In unstructured pruning, the unimportant filters or weights are set to zero, a compiler utilizes these zeros to skip some computations, thereby decreasing the inference time. Unstructured pruning has an additional cost in terms of compilation effort or computational effort in order to exploit the irregular sparsity. Typically, unstructured pruning provides a more pruning ratio as compared to structure pruning.

Network Pruning can also be classified in terms of *static pruning* and *dynamic pruning*. In static pruning, the parameters of a DNN are removed permanently, while in *dynamic pruning*, the parameters of a DNN are not removed permanently, instead, they are selectively used for computation based on the input to a DNN. A DNN can be pruned using static pruning, thereby reducing MACs and NPs, and then the same DNN can be dynamically pruned as well.

In [12], the authors surveyed dynamic neural networks. They divided dynamic networks into three main categories: (1) instance-wise dynamic models that process each input sample or instance with data-dependent architectures or parameters, (2) spatial-wise dynamic networks that conduct adaptive computation with respect to different spatial locations of image data, and 3) temporal-wise dynamic models that perform adaptive inference along the temporal dimension for sequential data such as videos and texts.

Our work comes under the category of instance-wise dynamic models. Within this category, the two types of dynamic networks that are relevant to our work are (1) dynamic depth models and (2) dynamic width models. In dynamic depth models, the sample could be predicted earlier in a network. Works such as [13] utilize this strategy. This strategy is based on the principle that an easier sample in a dataset can be predicted earlier within a network than a hard sample.

In contrast to dynamic depth models, the dynamic width models change the number of filters or channels of a CNN, based on the input sample. In [14], the authors utilized reinforcement learning for training an agent that judges the importance of each convolutional kernel and conducts channel-wise pruning conditioned on different samples such that the network is pruned more when the sample is easier. Their work utilizes hardness of sample for dynamic behavior in a DNN, whereas our work utilizes coarse prediction.

Similar to our work is [15], in which the authors proposed a Learning Kernel Activation Module (LKAM), which is able to dynamically activate or deactivate a subset of filter kernels depending upon the input image content during the inference phase. Their method requires a bank of 1x1 convolutional kernels followed by average pooling and a sigmoid function in order to choose which filter kernels in a layer will be activated.

Our work is unique in this regard that in addition to classifying easy samples earlier, we utilize an intermediate branch to perform a coarse prediction. The lower layers of a CNN output simpler features, while the higher layers output more categorical features that correlate with a specific class. In higher layers, different filters output features specific to various classes. The concept behind our proposed dynamic pruning is to perform a coarse prediction. Then based on this coarse prediction, we select the CNN filters only relevant to specific classes. explainable AI allows us to obtain filter importances relative to specific classes. Our approach allows us to reduce the average latency as well as the longest-path latency of inference while keeping the overhead low and hardware implementation easy.

## 1.1 CONTRIBUTIONS

Our contribution can be summarized in the following points:

- We propose a novel method for dynamic pruning that utilizes early exit along with early coarse prediction and explainable AI.
- The early coarse prediction branch is trained using deep top-k loss. If the branch predicts a sample with high confidence, the prediction is made. Otherwise, coarse prediction is obtained, thereby restricting the possible output to be within a subset of classes.
- The coarse prediction is used to dynamically select CNN filters relevant for those classes.

- The filters relevant for all output classes are obtained in prior using explainable AI. In run-time, only ranking of filters is done.
- The dynamically pruned model is trainable and easily deployable on the end hardware.

## 2 BACKGROUND

In this section, we briefly describe some of the concepts that are utilized in our work.

### 2.1 RANKING CRITERIA FOR PRUNING

One of the most essential tasks in filter pruning is to rank the filters based on their importances. The importance of a filter can be obtained locally within a layer or it can be obtained globally within a network. Some ranking criteria for pruning are as follows:

**Magnitude-based metrics**  These methods utilize $\ell_1$-norm and $\ell_2$-norm of the model weights and have been shown to work reasonably well for general cases in works such as [16] and [17].

**Loss-preservation based metrics**  These measures determine the effect of removing a set of parameters on model loss, for example, first-order Taylor decomposition has been used for this purpose by [18].

**XAI-based metrics**  As described in [19], *explainable AI* (XAI) seeks to explain why a neural network produces the output that it does for the input it gets. Explanations of DNNs can be of different types such as explaining which neurons are most sensitive (saliency methods), which neurons have the most effect on output, which input excites which neurons (signal methods), etc. For example, explainable AI has been used to guide the quantization and pruning of DNNs [20].

Saliency methods explain the decision of a neural network by assigning values that reflect the importance of input components in their contribution towards the output. These methods can be used to obtain the importance of both the features and the weights. Some of the methods in this category are for example: DeepLift [21], Conductance [22], IntegratedGradients [23], etc. We choose DeepLIFT as our work is not aimed at comparing different explainable-AI methods specifically, rather it utilizes them in a novel pruning strategy. We choose DeepLIFT due to its robustness and less computational requirement. One of the most attractive aspects of explainable AI based algorithms is that they provide scores of different filters respective to each output class, which makes it possible for these methods to be utilized in our dynamic pruning architecture.

**DeepLIFT**  In [21], the authors propose a method for decomposing the output prediction of a neural network on a specific input by backpropagating the contributions of all neurons in the network to every feature of the input.

DeepLIFT compares the activation of each neuron to its *reference activation* and assigns contribution scores according to the difference. The choice of reference activation is important for the algorithm's outcome, and it often requires domain-specific knowledge. To specify a reference activation, we must understand the intuition behind the DeepLIFT algorithm. It compares the effect of the features to a baseline of what the model would predict when it cannot see the features. Therefore, a good reference activation for MNIST [24] is an all-black image.

Mathematically, the DeepLIFT algorithm works as follows: Let $t$ represent some target output neuron of interest and let $x_1, x_2, \ldots, x_{n'}$ represent some neurons in some intermediate layer or set of layers that are necessary and sufficient to compute $t$. Let $t^0$ represent the reference activation of $t$. The quantity $\Delta t$ is defined as the difference-from-reference, that is $\Delta t = t - t^0$. DeepLIFT assigns contribution scores $C_{\Delta x_i \Delta t}$ to $\Delta x_i$ s.t.:

$$\sum_{i=1}^{n'} C_{\Delta x_i \Delta t} = \Delta t \tag{1}$$

$C_{\Delta x_i \Delta t}$ can be thought of as the amount of difference-from-reference $t$ that is attributed to the difference-from-reference of $x_i$. $\Delta t$ is the DeepLIFT score, which can be represented as follows:

$$\text{DL}(t,x) = \sum_{i=1}^{n'} C_{\Delta x_i \Delta t} \tag{2}$$

## 2.2 Obtaining filter importances from explainable AI algorithms

The explainable AI methods take a sample of a dataset as input and output sensitivity maps instead of activations/feature maps. These sensitivity maps have the same dimensions as the feature maps. Let $\Omega$ be a set of all indices of all feature map elements of all layers of a neural network, $I(m,\Omega)$ be the importances of all feature map elements from $m^{th}$ sample from the validation set, then we can define total feature map importance as the average of importances obtained from $M$ samples:

$$I(\theta) = \frac{1}{M} \sum_{m=0}^{M-1} I(m,\theta) \tag{3}$$

where $M$ is the number of samples used by the XAI method for obtaining the importances. $M$ varies depending upon the dataset, its typical value is 1–2% of validation samples for CIFAR10. These sensitivity maps are converted into the importances of the filter weights using different methods, which are as follows:

$\ell_1$-**norm criteria**    We can take $\ell_1$-norm of the sensitivity map. For example, for a $d \times d$ convolutional kernel, for each sample $m \in M$, the explainable AI method outputs a sensitivity map $O$ of height $H$ and width $W$, the sensitivity of the weights of this convolutional kernel is given by: $\sum_{w=0}^{W-1} \sum_{h=0}^{H-1} |O_{w,h}| \Big/ (H \times W)$.

**Max-Min criteria**    We can take also get importances by subtracting max from min of a importances, this is useful in DeepLIFT which also outputs negative importances to indicate pixels that negate a class. This will be given by : $\sum_{w=0}^{W-1} \sum_{h=0}^{H-1} \max(O_{w,h}) - \min(O_{w,h}) \Big/ (H \times W)$.

## 2.3 Early Exit of CNNs

In early exit CNNs, a number of exit blocks are placed with convolutional layers [13, 25]. Each exit block consists of a confidence branch and a prediction branch. The prediction branch makes the prediction and the confidence branch outputs the confidence score of the prediction. The branches utilize average pooling followed by a linear layer, which keeps the computational overhead low. The prediction branch utilizes the softmax function while the confidence branch utilizes the sigmoid function to generate the output.

For a neural network represented by parameters $\theta$, the output vector of the exit branch can be written as $y = F(x,\theta)$, where $x$ is the input image, $y$ is the input to a softmax function, and its output can be defined as:

$$\text{softmax}(y_i) = \frac{\exp(y_i)}{\sum_j \exp(y_j)}, \tag{4}$$

where $i$ indicates the index of the predicted class and $j$ indicates the index of other output classes. The output of the confidence branch can be defined as:

$$\sigma(y_i) = \frac{1}{1 + \exp(y_i)}, \tag{5}$$

if $\sigma(y_i)$ is more than a certain threshold, the model exits at the early exit branch. The threshold $t_c$ can be easily obtained using the validation dataset.

## 2.4 Deep top-k loss function

In addition to the early exit, we also perform a coarse prediction, and performing this optimally is important for our dynamic pruning approach. Formally speaking, we need to train the branch

using top-k classification. For this purpose, we utilize the loss function proposed in [26]. As the authors formulate the problem, the top-k classification can be performed with DNNs trained with the cross-entropy loss. For example, the state-of-the-art models trained with cross-entropy loss yield successful results for top-5 error, even though the cross-entropy loss is not tailored for the top-5 error minimization. However, in case of a limited amount of data or noisy data, minimizing with cross-entropy loss does not work as well.

Our use-case of performing top-k prediction in the intermediate layers of a DNN is similar in the sense that the input to the coarse prediction branch is noisy, based on which the branch has to be trained to perform top-k prediction. The authors take inspiration from multi-class Support Vector Machines (SVMs) and create a margin between correct top-k prediction and incorrect predictions. Another contribution of the work in [26] is to smoothen the loss with a temperature parameter. This allows smooth gradients and better training under the assumption of noisy data. The work can be followed for exact derivation of the deep top-k loss function.

## 3 PROPOSED ARCHITECTURE

A schematic of our proposed dynamic pruning architecture is shown in Figure 1. We now describe the components of our architecture.

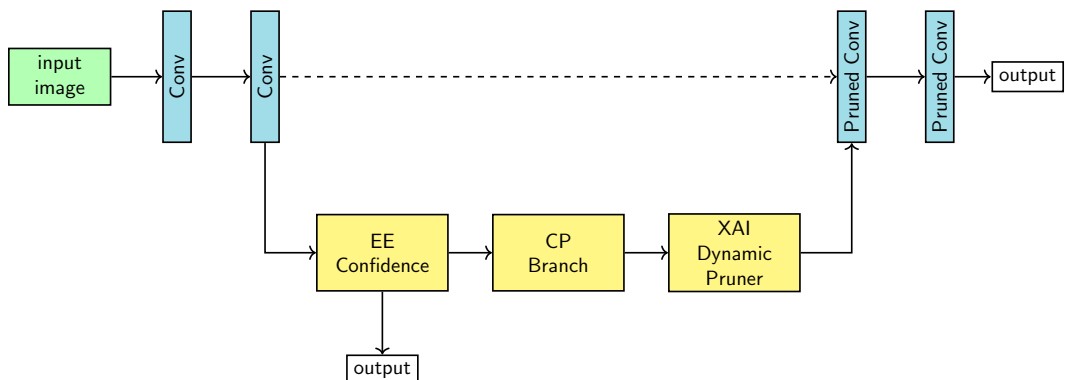

Figure 1: Schematic diagram of our dynamic pruning architecture.

### 3.1 EARLY EXIT (EE) CONFIDENCE

The EE confidence block consists of average pooling layer followed by a linear layer with a sigmoid function as activation. The goal of this branch is to provide a confidence measure to the top-1 prediction of the coarse prediction branch. If the top-1 prediction is made by the Coarse Prediction (CP) branch with high confidence, the sample exits the model and the prediction is made. As an example, for VGG11 on t the CIFAR10 dataset, 30% of the samples could be predicted with higher accuracy than the overall accuracy at the $5^{th}$ layer. So if the overall top-1 classification accuracy of the model is 90.8% and 30% of easier samples could be predicted at the $5^{th}$ layer with higher top-1 classification accuracy, then the prediction is made.

### 3.2 CP BRANCH

At CP branch, if the confidence of predicting the sample obtained from EE branch is greater than a threshold, the model is exited, otherwise, coarse prediction is performed. In order to train this branch, all the model parameters are frozen except the coarse prediction branch. The goal is to obtain maximum top-k accuracy in the coarse prediction branch, which can be used in the successive layers for selecting filters that are important for the coarse prediction. For example, with VGG11 on CIFAR10 dataset, we attach the coarse prediction branch after the $5^{th}$ layer and measure top-3 and top-5 accuracy. In our experiments, training the coarse prediction branch with deep-k loss function after setting $k = 1$ yields the best results. For VGG11 on CIFAR10, we train this branch for 10 epochs with a learning rate of 0.01 using Stochastic Gradient Descent (SGD) [27]. As shown in Table 1,

utilizing deep top-k loss function is significantly better for coarse prediction than the cross-entropy loss.

Table 1: Utilizing deep top-k loss function while setting $r = 1$ has significantly better performance than $r = 5$.

| method | top-5 accuracy | top-3 accuracy |
|---|---|---|
| deep top-k ($r = 1$) | 97.50% | 93.10% |
| deep top-k ($r = 5$) | 72.50% | 30.24% |
| cross entropy loss | 73.50% | 62.70% |

**Placing the coarse prediction branch**   Placing of the coarse prediction branch is dependent upon the constraints of accuracy, model, and the dataset. For example, for VGG11 model with CIFAR10 dataset, we placed the branch at the $5^{th}$ layer using a simple grid search. Branch is placed at each layer successively starting from the highest convolutional layer, trained for five epochs using deep top-k loss function and top-3 accuracy is obtained on the validation set. If the top-3 accuracy falls below a specific threshold, the branch is placed on layer ahead of it. We show the different top-3 accuracies obtained for different branch placements in Table 2.

Table 2: Top-3 accuracies for branch placement at different layers of VGG11.

| $7^{th}$ layer | $6^{th}$ layer | $5^{th}$ layer | $4^{th}$ layer |
|---|---|---|---|
| 95.3% | 94.4% | 93.1% | 85.7% |

### 3.3    XAI DYNAMIC PRUNER

This block does not contain any neurons. It stores the XAI-based importances of pruned layers. The importances are obtained once in the start and during prediction of the sample, the ranking of filters is done. Based on the coarse prediction, the XAI dynamic pruner selects the filter kernels of pruned convolution layers that need to be used. This block does the dynamic pruning.

Once the coarse prediction branch is trained, the network needs to be trained with dynamic pruning enabled. The model can be trained for an arbitrary amount of pruning ratio. For training purposes, the weights corresponding to the pruned filters are zeroes out, which allows the model to be differentiable and the training to occur. We name this phase *soft dynamic pruning* as the model thinning is not carried out yet. Whereas for hardware deployment, the *hard dynamic pruning*  is carried out, whereby the model is thinned.

**Hardware Deployment and post-deployment fine-tuning**    After the model has been trained using dynamic pruning, we proceed to convert the model into a hardware-deployable model. For hardware deployment, all that is needed is instead of zeroing the pruned filters, we instantiate convolutional layers with a fewer filter kernels and store the un-pruned kernels in buffer. Based on the coarse prediction, the XAI Dynamic Pruner selects a subset of un-pruned kernels from the buffer are uses them as weights of the convolutional layers. In terms of overhead, our method does not reduce the memory requirement of the model as compared to the un-pruned model. However, it reduces the latency by selectively utilizing the filter kernels. After this phase, the output layers of the model are fine-tuned.

## 4    EVALUATION

In this section, we evaluate our approach using standard datasets and commonly used deep learning models.

### 4.1    SOFTWARE AND HARDWARE

For all experiments, we used an NVIDIA Titan RTX GPU with 24 GB of memory and an Intel i7-9700 processor. The experiments were implemented using Python [28] as programming language in combination with the following software: PyTorch [29], numpy [30], matplotlib [31], scikit-learn [32], Captum [33], PyTorchCV [34].

## 4.2 METRICS

For evaluation, we utilize classification accuracy, average inference time ($t_{mean}$), longest-path inference time ($t_{long}$) and Floating Point Operations (FLOPs) (flops). $t_{mean}$ is calculated by measuring the average inference time over the entire test dataset, which includes samples that exit early. $t_{long}$ is the average inference time of the samples that do not exit early.

**Discussion of complexity:** We measure the additional complexity for implementing our dynamically pruned model in reference to the corresponding statically pruned model. We show that the computational complexity is negligible for most hardware platforms.

## 4.3 BASELINE AND REFERENCE

As reference, we use the un-pruned model in order to compare the speedup as well as accuracy. For the baseline, we train a statically pruned model with the same number of filters as the dynamically pruned model. The statically pruned model is pruned using one-shot pruning with $\ell_1$-norm of DeepLIFT as the ranking criteria. The statically pruned model is useful in giving us a measure of computational overhead due to the additional branches and processing that is incurred in dynamic pruning process.

## 4.4 VGG11 WITH CIFAR10

Firstly, we evaluate VGG11 with CIFAR10 dataset. VGG11 dataset consists of 11 layers, 8 of which are convolutional layers. The number of filters in these 8 layers are 64, 128, 256, 256, 512, 512, 512, and 512, respectively. We obtain a VGG11 model trained on CIFAR10 dataset. The pre-trained un-pruned reference model has a top-1 accuracy of 90.9% on the test dataset.

Then we proceed with our approach and first attach the early exit and coarse prediction branches and train these branches. In this configuration, the early exit and coarse prediction branch are placed after the $4^{th}$ layer. The early exit branch and coarse prediction branches are trained initially with 15 epochs with a learning rate of 0.01 and a momentum of 0.9 using SGD. This allows us to obtain a top-5 accuracy of around 97%, while 30% of samples can exit the network with an accuracy of 92.3%. After training, the coarse prediction and early exit branch, the soft dynamic pruning is turned on and the network is trained for 30 epochs. Then the network is converted into hardware deployable form and hard dynamic pruning is used. After hard dynamic pruning, only the final output layers of the network need to be fine-tuned for 15 epochs. The number of filters in the last four layers are reduced from 512 to 128. One could also employ layer sensitivities and iterative pruning, however our goal is to demonstrate the utility of our approach, so we pick one configuration for that purpose. The results for CPU are shown in Table 3.

Table 3: This table shows the performance of our architecture CPU with VGG11-CIFAR10 as the model-dataset pair.

| method | accuracy | $t_{long}$ | $t_{mean}$ | flops |
|---|---|---|---|---|
| Dynamic Pruning | 91.30% | 2.83ms | 2.65ms | 161.8M |
| Static Pruning Baseline | 89.40% | 2.70ms | 2.70ms | 159.5M |
| Reference Model | 90.09% | 5.60ms | 5.60ms | 171.9M |

**Results on CPU** As we can observe, the accuracy of our dynamically pruned model is even greater than the reference model, this happens when an over-parameterized model is pruned. Additionally, the static pruning baseline is useful in concluding that the computational overhead for introducing dynamic pruning is negligible. We can also observe that while the reduction in flops is not significant for pruned model, while the reduction in latency is almost double. This could be due to the reason that memory transfers take more time and have more influence on latency as compared to flops.

**Results on GPU** We obtain the results for GPU as well, which are shown in Table 4. The latency on GPU is reduced by a factor of 2, as we observed in the case of CPU as well.

Table 4: This table shows the performance of our architecture GPU with VGG11-CIFAR10 as the model-dataset pair.

| method | accuracy | $t_{long}$ | $t_{mean}$ | flops |
|---|---|---|---|---|
| Dynamic Pruning | 91.30% | 0.227ms | 0.211ms | 161.8M |
| Static Pruning Baseline | 89.40% | 0.220ms | 0.220ms | 159.5M |
| Reference Model | 90.09% | 0.447ms | 0.447ms | 171.9M |

## 4.5 RESNET20 WITH CIFAR10

We test ResNet20 with CIFAR10 dataset. This ResNet architecture consists of three different widths; six $3 \times 3$ convolutional layers consist of 16 filters, six other $3 \times 3$ convolutional layers consist of 32 filters and last six $3 \times 3$ convolutional layers consist of 64 filters, while two $1 \times 1$ convolutional layers are identity layers and one layer is a fully connected output layer. In this experiment, we prune the last six layers consisting of 64 layers and reduce them to 48. The results for measurements on CPU are shown in Table 5. As we can see, the overhead of dynamic pruning is low and the latency is close to the latency obtained for the static pruning baseline, where the accuracy for the dynamically pruned model is significantly higher.

Table 5: This table shows the performance comparison on CPU with ResNet20-CIFAR10 as the model-dataset pair.

| method | accuracy | $t_{long}$ | $t_{mean}$ | flops |
|---|---|---|---|---|
| Dynamic Pruning | 86.5% | 2.17ms | 2.13ms | 34.5M |
| Static Pruning Baseline | 78.5% | 2.12ms | 2.12ms | 34.0M |
| Reference Model | 94.3% | 2.45ms | 2.45ms | 40.5M |

## 4.6 VGG16 WITH CIFAR100

We test VGG16 with CIFAR100 dataset to show the application of our approach on a dataset with large number of classes as the CIFAR100 dataset contains hundred output classes. VGG16 model consists of 13 convolutional layyers, one with 64 filters, three with 128 filters, three with 256 filters and six with 512 filters. The exit branch is placed after the $7^{th}$ layer and last six convolutional layers are dynamically pruned by reducing the number of filters to 128, the top-5 coarse prediction is performed at the coarse prediction branch. The results are shown in Table 6.

Table 6: This table shows the performance comparison on CPU with VGG16-CIFAR100 as the model-dataset pair.

| method | accuracy | $t_{long}$ | $t_{mean}$ | flops |
|---|---|---|---|---|
| Dynamic Pruning | 71.2% | 8.14ms | 7.82ms | 440.0M |
| Reference Model | 71.4% | 13.20ms | 13.20ms | 666.3M |

## 4.7 COMPARISON WITH STAT-OF-THE-ART STATIC PRUNING

Our work is not directly comparable with the state-of-the-art static pruning approaches, as our dynamic pruning method allows for further prunability after the static pruning has been done. In order to demonstrate this, we utilize a VGG16 model that is statically pruned on CIFAR10 dataset using gate-decorator method proposed in [35]. The statically pruned model contains 17, 46, 57, 71, 73, 60, 32 filters in initial seven convolutional layers respectively and 51 filters each in the last six convolutional layers. We prune the last six layers dynamically and reduce the number of filters from 51 to 40. In terms of inference time, this gives us a speedup of 1.33 on CPU.

Table 7: This table shows the performance comparison on CPU between statically pruned VGG16 model for CIFAR10 dataset and dynamic pruning applied on that statically pruned model.

| method | accuracy | $t_{long}$ | $t_{mean}$ | flops |
|---|---|---|---|---|
| Dynamic Pruning | 91.2% | 2.67ms | 2.48ms | 199.9M |
| Reference Model | 91.3% | 3.33ms | 3.33ms | 209.75M |

## 5 Conclusion and Future Work

In conclusion, we propose a novel dynamic pruning architecture for pruning CNN filters that utilizes explainable AI and can be deployed on hardware with relative ease. It provides an impressive gain in terms of inference latency on different types of hardware. To the best of our knowledge, the idea of performing coarse prediction in intermediate layers and utilizing explainable AI to select filters important to those classes has not been proposed before. In comparison with existing static pruning approaches, our architecture has the advantage that it can be used after static filter pruning to obtain further speedup. In comparison with existing dynamic pruning approaches, our approach has the advantage of being able to reduce the longest-path inference time in addition to the average inference time.

For future work, some of the things could be improved further, for example, design space exploration for placement of the early exit and coarse prediction branches as well as efficient utilizing of multiple branches could be explored. The structure of the branches could also be changed from simpler linear layers to more complicated layers.

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
