# OpenReview forum: "EXPLAINABLE AI-BASED DYNAMIC FILTER PRUNING OF CONVOLUTIONAL NEURAL NETWORKS"
_ICLR.cc/2022/Conference — ICLR 2022 Submitted_

### Official Review · Reviewer_PWoD · 2021-10-20

**Correctness:** 3
**Technical Novelty And Significance:** 2
**Empirical Novelty And Significance:** 2
**Recommendation:** 3
**Confidence:** 3

**Main Review:**

1. Incomplete experimental results
1-1) Experimental results on more and more various datasets (e.g., ImageNet), networks (e.g., MobileNet, EfficientNet, ResNet50), and configurations (e.g., various FLOPs reduction ratio) should be included.
1-2) It is necessary to show whether the proposed method is effective not only for classification but also for more practical detection and segmentation networks.
1-3) As with other pruning-related SOTA papers, in order to prove the superiority of the proposed method, all experiments should be compared with various SOTA studies. Section 4.7 contains almost nothing.
1-4) Recently, activation functions with negative values such as Swish and Mish are widely used. In general, pruning is vulnerable to activation functions including negative values due to the influence of shift parameters, so please show whether the proposed method has robust properties to these activation functions.
1-5) As you know, fine-tuning requires a significant training cost, so many filter pruning techniques that achieve excellent performance without fine-tuning have been proposed. It is necessary to show whether good performance can be maintained even if fine-tuning is removed from the proposed method.

2. Novelty of the proposed method: It is difficult to find a clear novelty in the proposed method because it is mostly based on existing methods and only a few changes have been made.

3. It is difficult to agree with the argument of this paper because the motivation of this paper is not sufficiently presented. To compensate for the fact that the experimental results of this paper are not sufficient to prove the superiority of the proposed method, the motivation of this study needs to be clearly presented at the beginning for enhancing the novelty of this paper.

4. To support the sentence, “Our proposed architecture for dynamic pruning can be deployed on different hardware platforms.”, analysis and experimental results for more diverse hardware (e.g., embedded GPU board, FPGA, ASIC) should be accompanied.

5. Recently, many kernel pruning studies have been attempted, and therefore it would be good to explain the advantages of the proposed method compared to kernel pruning targeting the kernel in the filter.

6. The relationship and role of explainable AI in the proposed method are not well explained, and it is not clear why the authors performed studies belonging to instance-wise dynamic models among the three categories.

**Summary Of The Paper:**

This paper proposed a new dynamic filter pruning method that utilizes explainable AI along with early coarse prediction in the intermediate layers of a CNN. The early coarse prediction branch is trained using deep top-k loss, and the coarse prediction is used to dynamically select CNN filters relevant for those classes. The filters relevant for all output classes are obtained in prior using explainable AI. The dynamically pruned model by the proposed method is trainable and easily deployable on the various end hardware.

**Summary Of The Review:**

The contribution needs to be made clearer. In addition, the experiment part that supports the contribution needs to be thoroughly supplemented. Please address my concerns in "Main Review" through the rebuttal process.

---

### Official Review · Reviewer_nAuM · 2021-10-29

**Correctness:** 3
**Technical Novelty And Significance:** 2
**Empirical Novelty And Significance:** 3
**Recommendation:** 5
**Confidence:** 4

**Main Review:**

strengths:

good readability except for the XAI section  (which is to brief)
Easy to understand concept
faster on CIFAR-100

weaknesses:

-incremental novelty, it is a combination of existing ideas
-no speed gains on CIFAR-10
-experimental evidence is done using only two networks and only CIFAR datasets.

-What is needed: add a second modern network (post-resnet)
add a more complex dataset than just the sketchy cifar-10 datasets, e.g. LSUN, COCO or the smaller Pascal VOC or similar whatever size fits the budget, as long as the images are of a higher resolution than 32x32 have a higher intraclass variability than CIFAR.

Table 1 shows what ? top-k performance for the top-k prediction heads ?
- analyze the impact of the k in top-k  on the performance of the whole model after pruning.

The paper would strongly benefit from more extensive experiments.

Some missing details (weaknesses):

-section on pruning XAI is somewhat too brief and needs a clearer explanation of the process

They store relevant filters precomputed for each class  ?
Or on what outputs else are the deeplift scores initialized?

-"Based on the coarse prediction, the XAI dynamic pruner selects the filter kernels of pruned
convolution layers that need to be used."

How do they combine filters for the classes from the top-k prediction for k>1?
do they use only k=1?

-How much do you prune ? How do you decide how much to prune?
This is even more so important because for some reason CIFAR-100 (same resolution) results in a much larger+slower model (Table 6 vs Table 7). Is it just because of a 10x increased number of output classes and thus retained filters in the oversized linear branches of VGG-16?

language (minor):
- " The
importances are obtained once in the start and during prediction of the sample, the ranking of filters
is done. "
This sentence can be misread as:
The importances are obtained once in the start and during prediction of the sample.

It needs an ",and," or, better, a rewrite.

Why table 7 lists reference model when the text states a statically pruned baseline?

Question: with what loss and labels the confidence in the exit branch is trained?


Citations:
consider to cite https://ieeexplore.ieee.org/document/9502445 instead of the arxiv for [20]
also please consider to cite:
https://www.sciencedirect.com/science/article/pii/S0031320321000868
https://arxiv.org/abs/2103.06460


**Summary Of The Paper:**

The authors propose a set of steps in order to reduce the latency and the flops computation effort of neural networks. They add early exit prediction layers, a top-k prediction layer and dynamic pruning based on the set of classes from the top-k prediction layer. The pruning uses statistics precomputed from a validation dataset in order to decide which filters are to be used for which class. The statistics are obtained from deeplift scores.

**Summary Of The Review:**

Its a paper which is a bit below the perceived acceptance level. It is an empirical paper, therefore for a high ranked conference it should come with a larger experimental evaluation and clearer details on how the XAI was used and how much was pruned. Most problems have higher complexity than CIFAR-10/100.

---

### Official Review · Reviewer_vKUc · 2021-10-31

**Correctness:** 2
**Technical Novelty And Significance:** 1
**Empirical Novelty And Significance:** 1
**Recommendation:** 3
**Confidence:** 5

**Main Review:**

Weakness:

1. This work was evaluated on CIFAR 10 and 100. Without evaluation on a larger dataset (like ImageNet), the usefulness of this work is harder to evaluate?
2. This work also does not compare against strong alternatives like prior work in dynamic neural network pruning (eg- See [A] and [B] for relevant related work) or compare against state of the art static pruning techniques (See [C]) or more recent efficient architectures like MobileNet or EfficientNet.  The one static pruning technique compared against is L1 pruning using a one-shot framework. This is a really weak benchmark to compare against. Without comparison against these alternatives, its really hard to evaluate the effectiveness of this technique.
3. The motivations of using XAI for pruning criterion is also unclear, given that the added understanding of importance was not used anywhere during the work
4. The authors also dont motivate the reason for importance of reduction of longest-path latency of inference via discussing and showing this element to be a limitation of various prior work. Further, not all dynamic inference techniques suffer from this issue (eg - See [A]).


Questions for the author:

1. What is the importance of XAI based filter importance criterion? Wouldn't other type of filter pruning criterion also help with this objective?
2. One of the arguments that the authors make is that this work reduces the longest-path latency of inference, however the author do not provide any data showing what the worst case scenario for prior works looks like and why is that a problem worth solving?

Suggestions for improvement:
1. The authors should strengthen the motivation of this work (See weakness 3 & 4). There are a lot of open questions regarding the choices for the architecture made in this work that are not backed by specific data showing that the reasons for the said decisions are reasonable.
2. There is a huge body of work in the domain of dynamic filter pruning. See related work in [A] and [B]. Some of these work eliminate the need of optimizing for worst case latency and provide strong benchmarks to compare against

[A] https://arxiv.org/abs/1810.05331
[B] https://dl.acm.org/doi/10.1145/3417313.3429380
[C] https://arxiv.org/abs/2003.03033

**Summary Of The Paper:**

This paper focusses on a dynamic filter pruning technique that reduces the longest path latency of inference while using explainable AI (XAI) to help with determining pruning criterion. The approach uses an early coarse prediction branch that is used to perform a top-k classification. This branch is added to the middle section of the neural network. If this branch either predicts the output class with high prediction, then the rest of the computation is skipped. Else, this branch predicts the top-k classes for this input and executes the rest of the part of the NN. While executing this part, only filters useful for predicting these classes are loaded during execution. To determine the set of filters useful for a certain set of classes, the paper uses XAI to determine importance criterion for these filters for each class. The authors evaluate their work on CIFAR 10 and CIFAR 100 benchmarks on VGG11&16 and ResNet20 architecture while comparing against a dense benchmark and a simple static pruning technique.

**Summary Of The Review:**

The motivations in this paper are weakly established. Further the results in this paper have not been compared against relevant benchmarks or on more established benchmarks. Given these issues, I recommend rejecting this paper.

---

### Official Review · Reviewer_kGTk · 2021-11-01

**Correctness:** 2
**Technical Novelty And Significance:** 1
**Empirical Novelty And Significance:** 1
**Recommendation:** 1
**Confidence:** 4

**Main Review:**

Overall, I think it is an interesting approach, but the novelty is minimal. The authors combine multiples techniques that already exists in the literature. Although the combination is done in a clever way, all techniques are barely explained, making the reader to go to the original papers to properly reproduce their approach.

Regarding to the experimental setup, too much work has to be done before publishing it. There is no comparison against other dynamic pruning algorithms like [1] or [2]. Furthermore, it is only tested on huge networks that have a bad accuracy vs. size ratio. I suggest the authors to include both more datasets and networks.

Finally, the results obtained are not impressive if we compare them against the papers mentioned above.


[1] Liu, Z., Xu, J., Peng, X., & Xiong, R. (2018, December). Frequency-domain dynamic pruning for convolutional neural networks. In Proceedings of the 32nd International Conference on Neural Information Processing Systems (pp. 1051-1061).

[2] Lin, S., Ji, R., Li, Y., Wu, Y., Huang, F., & Zhang, B. (2018, July). Accelerating Convolutional Networks via Global & Dynamic Filter Pruning. In IJCAI (Vol. 2, No. 7, p. 8).

**Summary Of The Paper:**

The authors propose a dynamic pruning algorithm by combining multiple methods like early exit and a top-k loss function. The experimental results show a good performance over the vgg network, whereas the performance when using resnet architectures decays a little bit.

**Summary Of The Review:**

Although the idea is interesting, the novelty is minimal. Besides that, more work has to be done in the experimental section to prove this new algorithm can achieve state-of-the-art results in dynamic pruning.

---

### Decision · Program_Chairs · 2022-01-20

**Decision:**

Reject

**Comment:**

### Summary

This paper presents a technique to reduce the worst-case latency of inference. The key idea is to use a combination of early exit and filter selection to achieve its results. The filter selection predicts the top-k classes for the input and, using that indication, uses the filters that are the most relevant  (using DeepLIFT) to refine the result.

### Strengths (from Discussion)

- The idea is interesting. Early exit, mixtures of experts (one potential interpretation of the filter selection here), as well as pruning are interesting mechanisms for neural network efficiency.
There may be new opportunities to find synergies in their combination.

### Weaknesses (from Discussion)

- The clarity of writing could be significantly improved, particularly in the description and illustration of the constituent techniques. Figures, such as those in https://arxiv.org/abs/2008.13006, that clearly present the constitution of various layers, in particular, would help.

- There are relevant and applicable baselines that a comparison would contextualize the strength of the approach (as per Reviewer vKUc examples)

- ImageNet experiments appear to be within reach of this experimental apparatus (i.e., without extreme cost). Hence, such experiments would validate the applicability of this approach to practice.

- A small point arose that longest-path inference was not motivated. Work on optimizing tail latency (https://research.google/pubs/pub40801/) may be helpful contextualization here.

### Recommendation

My recommendation is Reject. The work here is a very promising start for a new idea. Though requests for additional experimentation and baselines can be ill-defined recommendations. Here, scaling of the results to ImageNet as well as comparing against baselines in the literature (as per Reviewer vKUc's examples) would provide much stronger scoping for this work.